

**Manure nitrogen production and application in cropland and rangeland during 1860 - 2014: A 5-minute gridded global data set for Earth system modeling**

Bowen Zhang[1], Hanqin Tian[1], Chaoqun Lu[2,1], Shree R. S. Dangal[1], Jia Yang[1], Shufen Pan[1,3]

[1]International Center for Climate and Global Change Research, Auburn University, Auburn, AL, 36849, USA; [2]Department of Ecology, Evolution, and Organismal Biology, Iowa State University, IA 50011, USA; [3]State Key Laboratory of Urban and Regional Ecology, Research Center for Eco-Environmental Sciences, Chinese Academy of Sciences, Beijing 100085, China

Corresponding author:

Dr. Hanqin Tian, e-mail: tianhan@auburn.edu




## Abstract

Given the important role of nitrogen input from livestock system in the terrestrial nutrient cycles and the atmospheric chemical composition, it is vital to have a robust estimation of the magnitude, spatiotemporal variation of manure nitrogen production and the application to cropland and rangeland across the globe. In this study, we used the dataset from Global Livestock Impact Mapping System (GLIMS) in conjunction with country-specific annual livestock population to reconstruct the manure nitrogen production from 1860 to 2014. The estimated manure nitrogen production increased from 21.4 Tg N yr$^{-1}$ in 1860 to 131.0 Tg N yr$^{-1}$ in 2014, with a significant increasing trend during 1860-2014 (0.7 Tg N yr$^{-1}$, $p < 0.01$). Changes in manure nitrogen production exhibited highly spatial variability and concentrated in several hotspots (e.g., Western Europe, India, Northeast China and Southeast Australia) across the globe over the study period. In the 1860s, northern mid-latitude accounted for ~ 52% of the global total manure production, while tropical region became the largest share (~ 48%) in the recent five years (2010-2014). Among all the continents, Asia accounted for over one-fourth of the global manure production during 1860-2014. Cattle dominated the manure nitrogen production and contributed ~ 44% of the total manure nitrogen production in 2014, followed by goat, sheep, chicken and swine. The manure nitrogen production applied to cropland and rangeland accounts for less than one-fifth of the total manure nitrogen production over the study period. The 5-arc minute gridded global data set of manure nitrogen production generated from this study could be used as an input for global or regional land surface/ecosystem models to evaluate the impacts of manure nitrogen on key biogeochemical processes and water quality, and the best management practices of manure nitrogen applications to cropland and rangeland across the globe could be important for food security and environmental sustainability. Datasets available at: https://doi.org/10.1594/PANGAEA.871980



## 1. Introduction


Human induced nitrogen flow, mainly driven by the increasing needs for food
production, had a tremendous impact on the earth's biogeochemical cycles (Bouwman et al.,
2013; Galloway et al., 2008; Liu et al., 2010). Chemical fertilizer use began to play an important
role in enhancing the crop yield since the 1960s (Lu and Tian, 2017; Potter et al., 2010); while
manure has long been recognized as a traditional source of soil nutrient for centuries and
contributed ~37% - 61% of the total nitrogen input to the land surface (Bouwman et al., 2013).
The manure production is expected to increase in the coming decades, due to the growing
demand for livestock population as a result of the ever-increasing human population and the
shifts in diet structure with more meat consumption (Herrero and Thornton, 2013). The resultant
changes have been suggested to surpass sustainability threshold (Pelletier and Tyedmers, 2010),
with substantial impact on biogeochemical processes in the terrestrial ecosystems (Tian et al.,

2016).

Growing application of manure nutrient has contributed to an increase in crop production,
and at the same time, has been identified as one of the major causes for a litany of environmental
problems which impinge on the land, aquatic ecosystem, and even the atmospheric composition
(Davidson and Kanter, 2014). To maintain high yield, farmers tend to apply large amounts of
nitrogen fertilizer and organic manure, especially in the intensively crop-producing system. A
recent study revealed that only 38% of total reactive nitrogen input were finally transferred into
harvested crop yield (Liu et al., 2016). Part of surplus nitrogen can be accumulated in the soil
nitrogen pools. Around 2% of annual manure-nitrogen have been converted to nitrous oxide,
which is the largest anthropogenic stratospheric ozone-depleting substance and the third most
important anthropogenic greenhouse gas (Davidson, 2009; Davidson and Kanter, 2014; Tian et





al., 2016). It has been suggested that manure was the single largest source of the anthropogenic
emissions of nitrous oxide in the 2000s (Davidson, 2009; Davidson and Kanter, 2014; Syakila
and Kroeze, 2011). At the same time, manure also acted as the dominant source of ammonia
($NH_3$), which played a vital role in the formation of atmospheric particulate matter (PM), such as
PM2.5 and atmospheric nitrogen deposition (Behera et al., 2013; Sutton et al., 2013). Thus,
growing manure production could lead to an increase in $NH_3$ emission which impairs the public
and environmental health (Sutton et al., 2013). The rest of surplus nitrogen can leach through the
soil profile and contaminate groundwater in the form of nitrate (Ju et al., 2006). Excess nitrogen
together with phosphorous can stimulate the eutrophication of inland water (Conley et al., 2009),
be transported far away from original sources and exacerbate coastal water quality, and result in
hypoxia (Burkart and James, 1999; Yang et al., 2015). Environmental pollutions caused by
manure nitrogen enrichment have been reported all over the world, nonetheless, there are some
other places in the world, such as Africa, in which nitrogen scarcity still exists and even threaten
the food security (Liu et al., 2010; Zhou et al., 2014).
To determine the status of unevenly distributed nitrogen at large scales, it is critical to
have a good understanding of the geographic distribution of nitrogen inputs from each key
sector. In spite of extensive studies in the development of nitrogen fertilizer data at both regional
and global scales (FAOSTAT, 2014; Lu and Tian, 2017; Matthews, 1994; Nishina et al., 2016;
Potter et al., 2010), most previous datasets for manure nitrogen production at global scale either
relied on the livestock population dataset with coarse resolution, or were only available for
limited time periods without consistent inter-annual variation, e.g., Herrero and Thornton,
(2013), Holland et al., (2005), Liu et al., (2010), and Potter et al., (2010). Recent research has
expanded the estimation of manure nutrient production in the conterminous United States during



1930 - 2012 and in China to a period 2002 to 2008 (Ouyang et al., 2013; Yang et al., 2016). In
the conterminous United States, the manure nitrogen has increased by 46% from 1930 to 2012,
with substantial spatial heterogeneity (Yang et al., 2016). In China, manure nutrients are
unevenly distributed, with seven provinces contributing over half of the total manure nitrogen
(Ouyang et al., 2013).

Although these datasets have expanded our recognition of manure nutrient estimate,

spatially explicit estimates of manure nitrogen production are still lacking. To reduce the
uncertainty in estimating several key biogeochemical processes at the global scale, such as the
continuously increased emission of nitrous oxide, the occurrences of inland and coastal hypoxia
due to nutrient enrichment at large scales, it is necessary to identify the spatial and temporal
variation of manure nitrogen production over a long time period. Together with other data,
quantification of manure nutrient production could also be used to generate the comprehensive
assessment for livestock sectors and design sustainable options for the sector's development
(Herrero and Thornton, 2013). At the same time, it could quantify the uncertainties of analyzing
the key nutrient cycles in terrestrial ecosystems and its feedback to the climate over a century-
long period. The development of the Global Livestock Impact Mapping System (GLIMS) offers
an exceptional opportunity to improve manure data upon earlier studies and extend our
knowledge of manure production over a century-long period. Thus, the major objective of this
study is to produce global gridded maps at 5-arc minute resolution in latitude by longitude of
manure nitrogen production since 1860. More specifically, we (1) estimate the magnitude, spatial
and temporal variation of manure nitrogen production, (2) quantify the relative contribution of
major livestock groups on the manure nitrogen production, (3) investigate the spatial and

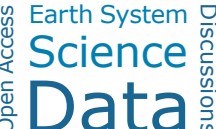

temporal variation of manure nitrogen applied to cropland and rangeland, and (4) discuss the
impacts of manure nitrogen production on the terrestrial biogeochemical cycles.



## 2. Method

### 2.1 Manure Nitrogen Production

To develop the gridded annual nitrogen production rate from manure during 1860-2014, we used the dataset from GLIMS, which provided the information of the spatial distribution of different livestock at a spatial resolution of 0.00833 degrees (- a nominal pixel resolution of approximately 1km×1km at the equator) for cattle (dairy and other cattle), swine, chickens, goats, sheep and a partial distribution for ducks (Robinson et al., 2014) (http://www.livestock.geo-wiki.org/). The annual variation of global livestock stock from 1961 to 2014 was controlled by country-specific information from FAOSTAT (FAOSTAT, 2014) (http://faostat.fao.org/site/291/default.aspx). For the countries (including United States, Australia, Brazil, Canada, China and Mongolia) with sub-regional (province/state level) livestock populations, we disaggregated FAO country level livestock population into sub-regions (see detailed description of the datasets in Dangal et al. (2017)). For those years without the information of livestock populations from FAOSTAT, we applied the annual trend extracted from the HYDE (History Database of the Global Environment, http://themasites.pbl.nl/tridion/en/themasites/hyde/landusedata/livestock/index-2.html) – livestock populations based on the continental analysis to fill the gaps. Default values for nitrogen excretion rate of different animals from Intergovernmental Panel on Climate Change (IPCC) 2006 guidelines (Tier1) were used. The trend of the inter-annual variation for manure nitrogen production before 1960 was obtained from Holland et al. (2005), and was applied to each grid to estimate the amount of manure nitrogen production from 1860 to 1960 (**Fig. 1**).

The development of the time-series nitrogen excretion rate from livestock is provided below in more details. To distribute the yearly country-level livestock population from



FAOSTAT, we standardized the livestock distribution with spatially explicit gridded information
from GLIMS to match the annual country-level livestock records from FAOSTAT.

$$D(FAO)_{i,j,k} = D(GLIMS)_{i,j} \times \frac{NTH(FAO)_{i,j,k}}{NTH(GLIMS)_{i,j}} \qquad (1)$$

Where: $NTH$ indicated the national total head of animal $j$ from a specific country $i$ (unit: head) in
year $k$. $D$ indicated the density of animal $j$ from a specific country $i$ (unit: head km$^{-2}$ land in each
grid) in year $k$.
Then we calculated the average nitrogen excretion rate by applying the IPCC 2006 guidelines
(Tier1).

$$Nex_{(i,j)} = N_{rate(i,j)} \times \frac{TAM_{(i,j)}}{1000} \times 365 \qquad (2)$$

Where $Nex_{(i,j)}$ indicated annual N excretion for livestock category $j$ from a specific country $i$
(Unit: kg N animal$^{-1}$ yr$^{-1}$); $N_{rate(i,j)}$ indicated default N excretion rate for livestock category $j$
from a specific country $i$ (Unit: kg N (1000 kg animal mass)$^{-1}$ day$^{-1}$); $TAM_{(i,j)}$ indicated typical
animal mass for livestock category $j$ from a specific country $i$ (Unit: kg animal$^{-1}$). Here, we
obtained country-level N excretion rate of livestock in China from Ouyang et al. (2013) and in
the conterminous United States from Yang et al. (2016). For other places in the world, we used
the N excretion rate for livestock from IPCC 2006 guidelines.
We calculated the gridded average nitrogen excretion rate.

$$Nman_{(i,j,k)} = Nex_{(i,j)} \times D(FAO)_{(i,j,k)} \qquad (3)$$

Where $N_{man(i,j,k)}$ indicated gridded average nitrogen excretion rates for livestock category $j$
from a specific country $i$ in year $k$ (Unit: kg N km$^{-2}$ yr$^{-1}$).
**2.2 Manure Nitrogen Applied to Cropland and Rangeland**



We further developed the gridded map of the manure nitrogen applied to cropland and
rangeland at 5-arc min resolution based on manure management in three livestock production
systems, including rangeland-based systems, mixed rainfed farming system, and mixed irrigated
farming system for cattle (dairy and other cattle), goat and sheep, and smallholder and industrial
systems for poultry and swine (Herrero et al., 2013). The livestock systems were further
classified according to the agroecological differentiations (arid-semiarid, humid-subhumid, and
temperate/tropical highland areas). Thus

$$F_{M(j,ProSys)} = F_{MT(j,ProSys)} * \left(1 - F_{MO(j,ProSys)}\right) * \left(1 - F_{Loss(j,ProSys)}\right) \qquad (4)$$

Where $F_{M(j,ProSys)}$ indicated the fraction of manure from livestock category $j$ applied to cropland
and rangeland, $F_{MT(j,ProSys)}$ indicated the fraction of total manure managed for different
livestock production systems. $F_{MO(j,ProSys)}$ indicated the fraction of managed manure to other
use, e.g., production of biogas. $F_{Loss(j,ProSys)}$ indicated the fraction of managed manure lost
through volatilization as $NH_3$ and $NO_x$. $ProSys$ indicated livestock production systems for cattle
(dairy and other cattle) and small ruminants, including rangeland-based systems (LGY:
Livestock-only systems in HyperArid areas, LGA: Livestock-only systems in Arid areas, LGH:
Livestock-only systems in Humid areas, and LGT: Livestock -only systems in Temperate areas
or Tropical Highland), Mixed rainfed farming systems (MRY: Mixed rainfed systems in
HyperArid areas, MRA: Mixed rainfed systems in Arid areas, MRH: Mixed rainfed systems in
Humid areas, MRT: Mixed rainfed systems in Temperate areas or Tropical Highlands), and
Mixed irrigated farming systems (MIY: Mixed irrigated systems in HyperArid areas, MIA:
Mixed irrigated systems in Arid areas, MIH: Mixed irrigated systems in Humid areas, and MIT:
Mixed irrigated systems in Temperate areas or Tropical Highlands), and Smallholder (POsm)
and Industrial (POin) for poultry and swine. The data of spatial distribution for livestock



production systems for ruminants, swine and chicken were obtained from GLIMS
(http://www.livestock.geo-wiki.org/download/), which represents the status around 2006.

To develop the spatial maps for manure nitrogen applied to soils in cropland and

rangeland during 1860-2014, we made several assumptions due to absence of the appropriate
data and calculated as

$$Nman_{CR(j,k)} = Nman_{(i,j,k)} \times$$

$$
\begin{cases}
F_{M(j,ProSys_{rl})} & a \\
F_{M(j,ProSys_{rd})} \times \frac{f_{crp(k)}}{f_{crp(2006)}} + F_{M(j,ProSys_{rl})} \times \left(1 - \frac{f_{crp(k)}}{f_{crp(2006)}}\right) & b \\
F_{M(j,ProSys_{irri})} \times \frac{f_{irri(k)}}{f_{irri(2006)}} + \left\{F_{M(j,ProSys_{rd})} \times \frac{f_{crp(k)}}{f_{crp(2006)}} + F_{M(j,ProSys_{rl})} \times \left(1 - \frac{f_{crp(k)}}{f_{crp(2006)}}\right)\right\} \times \left(1 - \frac{f_{irri(k)}}{f_{irri(2006)}}\right) & c
\end{cases}
$$

(5)

Where $F_{M(j,ProSys_{rd})}$ indicated the fraction of manure applied to mixed rainfed farming systems,
including MRY, MRA, MRH and MRT. $F_{M(j,ProSys_{irri})}$ indicated the fraction of manure applied
to mixed irrigated farming systems, including MIY, MIA, MIH and MIT. $F_{M(j,ProSys_{rl})}$ indicated
the fraction of manure applied to rangeland-based systems. $f_{irri(k)}$ indicated the fraction of
irrigated area to the total area in year $k$ in each grid cell. $f_{crp(k)}$ indicated the fraction of cropland
area to the total area in year $k$ in each grid cell.

The spatial distribution of livestock production systems in 2006 serves as a baseline map

to characterize the change of livestock production system during 1860-2005. We assumed the
spatial distribution of livestock production system remained the same during 2006-2014. We
assumed if the grid cell was identified as rangeland-based systems, the livestock production
system remained the same during the study period (See Eq. (5)-a); if the grid cell was identified
as mixed rainfed farming systems, the percent change of livestock production system would be
proportional to the changes of the cropland area in that grid cell backward from 2006, and the





mixed rainfed farming systems was converted from rangeland-based system (See Eq. (5)-b); if
the grid cell was identified as mixed irrigated farming systems, the percent change of livestock
production system would be proportional to the changes in the irrigated area in that grid cell
backward from 2006 and the mixed irrigated farming systems was converted from mixed rainfed
farming systems  (See Eq. (5)-c).

The gridded cropland distribution map during 1860-2014 was obtained from History

Database of the Global Environment version 3.2 (HYDE 3.2). We spatialized the country-level
actual area equipped for irrigation from FAOSTAT during 1961-2014 by adopting the gridded
irrigated area (expressed as percentage of area equipped for irrigation) (Siebert et al., 2013), to
create the gridded irrigation map during 1961-2014. We assumed the irrigated area didn't change
before 1961.

We assumed if the grid cell was identified as smallholder for poultry and swine, the

livestock production system remained the same during the study period; if the grid cell was
identified as industrial, the fraction of industrial livestock production system was assumed to be
0 in 1860, and 1 in 2006, and linearly increase from 1860 to 2006 for swine and chicken.

Previous studies suggested that the intensive duck producing system first came out in the

early 1950s (Ahuja, 2013; Raud and Faure, 1994). Thus we assumed the intensive duck
production system was assumed to be 0 in 1950, and 81.6% in 2008 and linearly increase from
1950 to 2008. The rest was occupied by extensive duck production systems (Ahuja, 2013; Duc
and Long, 2008; MOA, 2013; Raud and Faure, 1994).





**3. Result**
3.1 Temporal changes in manure nitrogen production
In this study, we quantified the total manure nitrogen production from six livestock
categories, including cattle (dairy and other cattle), chicken, duck, goat, swine, and sheep at a
global scale during 1860-2014 **(Fig. 2)**. We referred to the total mass of nitrogen excreted by
livestock for the manure nitrogen production. The estimated global manure nitrogen production
increased about 5 times, from 21.4 Tg N yr$^{-1}$ in 1860 to 131.0 Tg N yr$^{-1}$ in 2014, with an overall
significant increasing trend during 1860-2014 (0.7 Tg N yr$^{-1}$, $p < 0.01$) **(Fig. 3)**. In 1990, there
was near peak manure production (~116.3 Tg N yr$^{-1}$) followed by a decrease to 1998 (108.4 Tg
N yr$^{-1}$) and then increase again. The global manure nitrogen production rate increased from 146.2
kg N km$^{-2}$ yr$^{-1}$ in 1860 to 839.4 kg N km$^{-2}$ yr$^{-1}$ in 2014.
3.2 Spatial patterns of manure nitrogen production
Manure nitrogen production exhibited a large spatial variability over the study period. In
the 1860s, northern mid-latitude (NM, 30°N-60°N) accounted for over half of the global total
manure production (~12.0±0.5 Tg N yr$^{-1}$, Avg. ± 1 std. dev., same hereafter). Tropical regions
(30°N-30°S) contributed another one-third of total manure nitrogen production, followed by
Southern mid-latitude (SM, 30°S-60°S) (~12.7%), and the northern high-latitude (NH, 60°N-90°N)
(~0.8%). However, the dominant regions of the total manure nitrogen production have been
changed in the recent years. During the recent 5 years (2010-2014), tropical region took the largest
share, which was around 48.0% of the estimated global manure production (~61.9±0.9 Tg N yr$^{-1}$),
followed by NM (~37.7%), SM (~13.9%), and NH contributed the least part of the global manure
nitrogen production **(Fig. 4)**.



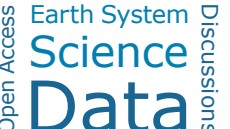

From the continental perspective, manure nitrogen production in Europe (~6.2±0.3 Tg N
yr$^{-1}$) appeared to be similar as that in Asia (~6.0±0.2 Tg N yr$^{-1}$) in the 1860s, which was much
higher than that in any other continent, including South America (~3.6±0.1 Tg N yr$^{-1}$), Africa
(~2.8±0.1 Tg N yr$^{-1}$), North America (~2.6±0.1 Tg N yr$^{-1}$) and Oceania (~1.9±0.1 Tg N yr$^{-1}$).
During 2010-2014, however, Asia accounted for the largest single share (~34.2%), followed by
Africa (~17.6%), South America (~14.2%), Oceania (~13.3%), Europe (~11.6%), and North
America (~9.2%) **(Table 1).**
Despite an overall increasing trend in all the continents, changes in manure nitrogen
production showed highly spatial variability and revealed several hotspots over the globe due to
the imbalances of global economic development and population growth **(Fig. 5)**. Southern Mexico,
Central America, Columbia, Southern Brazil, Uruguay, Western Europe, India, Northeast China
and Southeast Australia experienced a significant increase in manure nitrogen production during
1860-2014. For instance, the annual increase rate of manure nitrogen production in India (~11.0
kg N km$^{-2}$ yr$^{-1}$) was over two times higher than that at global scale (~kg N km$^{-2}$ yr$^{-1}$) during the
last 155 years.
3.3 Relative contribution of different livestock categories
At global level, among different livestock categories cattle dominated the manure nitrogen
production and contributed around 55.5% and 43.7% of the total manure nitrogen production in
1860 and 2014, respectively. Goat and sheep together contributed another one third of the total
manure nitrogen production during the study period, followed by swine and chicken. Duck shared
the least portion of manure nitrogen production. However, at regional level identifying the
dominant livestock species to the total manure nitrogen production, duck was the dominant
contributor in Alaska and Canada, while cattle played a dominant role in the conterminous United



States, Mexico, India, and most areas in South America and Europe **(Fig. 6)**. Goat contributed the
most in North Africa, Australia, and central and northeast Asia, while chicken and swine
dominated in Russia.
3.4 Spatial and temporal variation of manure nitrogen applied to cropland and rangeland

At global scale, the manure nitrogen applied to cropland and rangeland increased from 3.6

Tg N in 1860 to 24.5 Tg N in 2014, with a significant increasing trend (1.4 Tg N decade$^{-1}$, $p<0.01$)
during 1860-2014. The application to cropland and rangeland only accounted for 16.9% and 19.1%
of the total manure nitrogen production over the study period. Among different livestock
categories, cattle (dairy cattle together with other cattle) contributed around half (42.4% ~58.7%)
of the total manure nitrogen applied to cropland and rangeland. Other ruminants (goat and sheep)
only accounted for another 14.5% ~ 22.1% over the study period, which was similar to the
contribution from swine (16.9% ~ 23.3%). At the continental scale, Europe was the dominant
contributor (27.8% ~ 37.3% of global total) before the 1990s; however, its manure production
reduced dramatically since the early 1990s **(Fig. 7)**. Asia accounted for 24.4% ~ 37.7% of the
global manure nitrogen applied to cropland and rangeland over the study period at the fastest
growing rate of 0.47 Tg N decade$^{-1}$ compared to other continents.



**4. Discussion**

4.1 Comparison with previous studies

Over the last two decades, due to the recognition of the importance of manure nitrogen production in the nitrogen cycles, various previous studies have estimated the manure nitrogen production at both regional and global levels. At the global scale, it has been suggested that manure nitrogen production increased from 26.3 Tg N yr$^{-1}$ in 1860 to 142.5 Tg N yr$^{-1}$ in 2004, with an increasing trend of 0.84 Tg N yr$^{-1}$ (Holland et al., 2005), which was 18.5% higher than our estimate from 1860 (~21.4 Tg N yr$^{-1}$) to 2004 (~119.1 Tg N yr$^{-1}$). However, our result during the 1990s (~110.0±1.9 Tg N yr$^{-1}$) was more consistent with estimates from other studies, ranging from 101.4 Tg N yr$^{-1}$ to 128.3 Tg N yr$^{-1}$ (Bouwman et al., 2009; Potter et al., 2010; Van der Hoek et al., 1999). There were some spatial differences between the estimated manure nitrogen production by this study and Bouwman et al., 2013 and Potter et al., 2010 **(Fig. 8)**, partly due to the difference in calculation processes. Bouwman's estimate for manure nitrogen applied to cropland and rangeland is higher than our estimate, mainly due to the consideration of more refined manure management in different livestock production systems from our study. Our analyses indicated that the total amount of manure production in different continents was close to other estimates, with the difference around ±4% (Difference = $\frac{\text{Estimate from this study} - \text{Estimate from Potter et al.,2010}}{\text{Estimate from this study}}$). Our results showed that manure nitrogen production in Europe started to decline since the early 1990s, which was mainly due to the reduction of livestock population in Europe (FAOSTAT, 2014). At country scale, our estimation of manure nitrogen production (~5.3±0.8 Tg N yr$^{-1}$) was close to the previous estimation for the conterminous United States (~5.9±0.7 Tg N yr$^{-1}$) during 1930–2012 (Yang et al., 2016). Meanwhile, both studies identified cattle as the dominant contributor for the manure nitrogen production in the conterminous United States. For the manure nitrogen applied





to cropland and grassland in China, our estimation (3.0~3.6 Tg N yr$^{-1}$) was lower than previous
studies (5.1 ~ 6.2 Tg N yr$^{-1}$) from 2002 to 2008 (Ouyang et al., 2013), which might be due to our
consideration of livestock-specific and region-specific manure management factors to calculate
the amount applied to cropland and rangeland.
4.2 Manure production in the context of global environmental changes

During the past 155 years, the nitrogen input from atmospheric deposition increased

smoothly, with a significant increasing rate of 0.36 Tg N yr$^{-1}$ (Dentener, 2006; Wei et al., 2014).
The nitrogen fertilizer use began to step into sights and altered the global nitrogen cycle since the
early 1960s. The fertilizer use has increased around 835% during the past six decades, with a
significant increasing trend of 1.8 Tg N yr$^{-1}$ (EPI, 2016). The magnitude of nitrogen production
from manure was always higher than fertilizer consumption (**Fig. 3**), despite only 16.9%~19.1%
of the total produced manure nitrogen could be applied to cropland and rangeland. The increasing
population of livestock driven by human demand alters the global nitrogen cycle, reduces the
nitrogen use efficiency, and regulates the international food and feed trade (Davis et al., 2015;
Lassaletta et al., 2014). Livestock manure played a dual role in soil nutrients (Sattari et al., 2016).
While acting as nitrogen input to soils, it was also a pathway for nitrogen leaving the systems since
manure was originated from cropland and/or rangeland, which provide feed for livestock. In some
countries the intensification of the livestock production systems was based on importing feed from
other places of the world, where the nutrients were extracted and exported far away from where
they were generated, which influence the imbalance of nitrogen status all over the world.

Previous studies suggested that manure nitrogen production is the single largest source of

nitrous oxide emission (Davidson, 2009; Davidson and Kanter, 2014). Here, we compared the
estimated global manure nitrogen production with atmospheric nitrous oxide mixing ratio after





1977, which was obtained from NOAA (http://www.esrl.noaa.gov/gmd/aggi/aggi.html). Before
1977, we used the atmospheric nitrous oxide mixing ratio from Machida et al. (1995). The
estimated global manure nitrogen production showed a significant correlation with atmospheric
nitrous oxide mixing ratio, with correlation coefficient of 0.9679. By using the regression equation
derived by Davidson (2009), we could roughly estimate that manure-induced $N_2O$ emission was
around 2.7 Tg $N_2O$-N $yr^{-1}$ in 2014, which accounted for 21.1% and 17.5% of the total biogenic
$N_2O$ emission estimated by top-down approach and bottom-up approach, respectively (Tian et al.,
2016). In addition to the contribution of greenhouse gases, the intensive and extensive livestock
husbandry has already introduced large amounts of non-point pollution into the inland water and
are suggested to extend the summer time of hypoxia in the Gulf of Mexico (Goolsby, 2000; Yang
et al., 2015).
4.3 Uncertainties
Our study estimates the magnitude and spatiotemporal distribution of manure nitrogen
production over the globe during 1860-2014. There are several uncertainties needed to be
considered while interpreting the results from this study. First, the estimation of manure nitrogen
production is based on country-level animal population data together with regional level and
livestock-specific excretion rate. However, the uniform excretion rate for specific livestock type
at regional scale could bring some uncertainties without considering the feed availability and
quality across different seasons and various regions (Ouyang et al., 2013; Rufino et al., 2014).
Second, we used one-phase static GLIMS to get the baseline map of livestock distribution, which
was not able to provide the accurate information of change in spatial distribution of livestock at
sub-national level over time. For instance, the free grazing livestock may migrate due to the
availability of the food especially in the beginning of the study period. Therefore, the spatial



distribution of different livestock at sub-national scale, such as cattle, sheep and goat, might be
different considering livestock migration. In addition, we made several other assumptions to
develop global datasets for manure nitrogen production and manure nitrogen applied to cropland
and rangeland due to the absence of appropriate dataset, which definitely could introduce some
uncertainties. When using this dataset for specific use, further analysis or assumption needs to be
made to fulfill the objectives from different studies (Yang et al., 2016).
4.4 Implication

Livestock manure has long been recognized to be a double-edge sword in most

agricultural systems (Potter et al., 2010). While providing manure for arable land and increasing
food production, it also leads to a series of environmental problems. Continuous intensification
of livestock system could increase the potential of concentrated manure nitrogen load at specific
regions and exaggerate the environmental problems if without effective systematic livestock
management practices (Ouyang et al., 2013; Yang et al., 2016). Recent studies indicated that
intensification of the livestock system mainly relied on imported feed disconnected from local
crop and lead to great loss of manure nitrogen as well as direct environmental pollution
(Lassaletta et al., 2014). However, if the integrated manure management systems are being
applied and most of the manure could be recycled and applied to the agricultural land and
partially substituted the need for the synthetic fertilizer use, it would result in around 12%
reduction of the global nitrogen surplus (Bouwman et al., 2013).

Moreover, to improve the understanding of manure nitrogen production at large scale,

more regional information is needed. For example, the detailed excretion rate for different
livestock groups at specific region over time could enhance the accuracy of current estimation,
especially in those hotspots region with intensive livestock growth, such as India. It also needs to





consider the impact of the social economic development, environmental factors and climate
change on the livestock population dynamics and livestock structures, which could also influence
the manure nutrient production.



## 5. Conclusion

The global nitrogen cycle has been remarkably altered by anthropogenic activities. Due to the increasing demand of food production and alteration of diet structure, the livestock population has increased dramatically and is expected to continue increasing in the future. In this study, we quantified the spatially explicit global manure nitrogen production across the globe during 1860-2014. The estimated total manure nitrogen production increased from $23.1 \pm 1.0$ Tg N $yr^{-1}$ in the 1860s to $128.8 \pm 1.3$ Tg N $yr^{-1}$ during the recent five years (2010-2014), with an overall significant annual increasing trend during 1860-2014 (0.7 Tg N $yr^{-1}$, $p < 0.01$). From the latitudinal band perspective, tropical and northern middle latitudes dominated the estimated global manure nitrogen production. From the continental perspective, Asia shared the largest portion of global manure nitrogen production during recent decades. Southern Mexico, Central America, Columbia, Southern Brazil, Uruguay, Western Europe, India, Northeast China and Southeast Australia increased most significantly in manure nitrogen production during 1860 - 2014. The manure nitrogen production applied to cropland and rangeland only accounted for 16.9% ~ 19.1% of the total manure nitrogen production over the study period. It is expected to comprehensively evaluate the tradeoff between food production, climate mitigation and environmental pollution caused by the application of manure to further improve manure management. Together with other data, this 5-arc minute gridded dataset could be used as input of ecosystem and earth system models for assessing the impact of manure production on global biogeochemical processes, water resource and climate change.



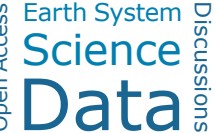

**Acknowledgements:** This study has been supported by NSF Dynamics of Coupled Natural and
Human Systems (1210360) and Chinese Academy of Science.





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





**Table 1 Estimation of manure nitrogen production at continental scale**

| Manure nitrogen production (Tg N yr$^{-1}$) | 1860s | 1900s | 1940s | 1980s | 2010s |
|---|---|---|---|---|---|
| Asia | 6.0±0.2 | 9.8±0.4 | 14.8±0.7 | 29.3±1.2 | 44.3±0.7 |
| North America | 2.6±0.1 | 4.2±0.2 | 6.3±0.3 | 10.7±0.2 | 11.8±0.04 |
| Europe | 6.2±0.3 | 10.1±0.4 | 15.3±0.7 | 25.7±0.1 | 14.9±0.1 |
| Africa | 2.8±0.1 | 4.6±0.2 | 6±0.4 | 13.3±0.6 | 22.6±0.8 |
| South America | 3.6±0.1 | 5.9±0.2 | 8.9±0.4 | 14.4±0.4 | 18.3±0.1 |
| Oceania | 1.9±0.1 | 3.1±0.1 | 4.7±0.2 | 13.0±5.4 | 17.2±0.1 |
| Global | 23.1±1.0 | 37.5±1.4 | 57.0±2.8 | 106.4±7.3 | 129.0±1.5 |







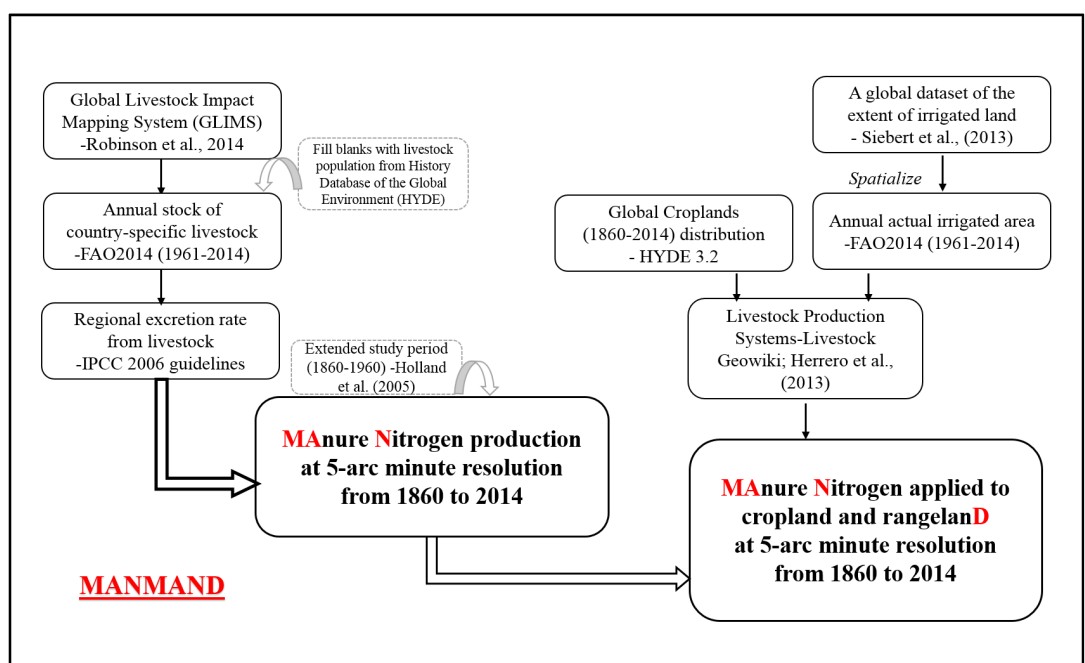



**Figure 1 Workflow for developing the global gridded data of manure nitrogen production and manure nitrogen applied to cropland and rangeland during 1860-2014**










**Figure 2 Spatial distribution of manure nitrogen production across the global land surface in the four years (1860, 1930, 1980 and 2014)**


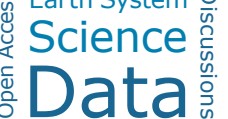



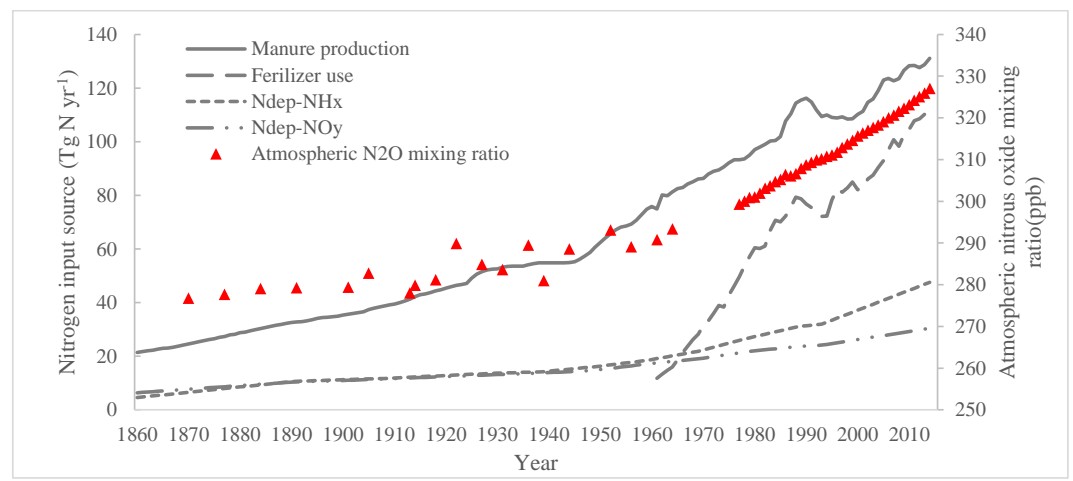


**Figure 3 Comparison of nitrogen input from global manure production, fertilizer use and atmospheric nitrogen deposition with atmospheric nitrous oxide mixing ratio during 1860-2014**





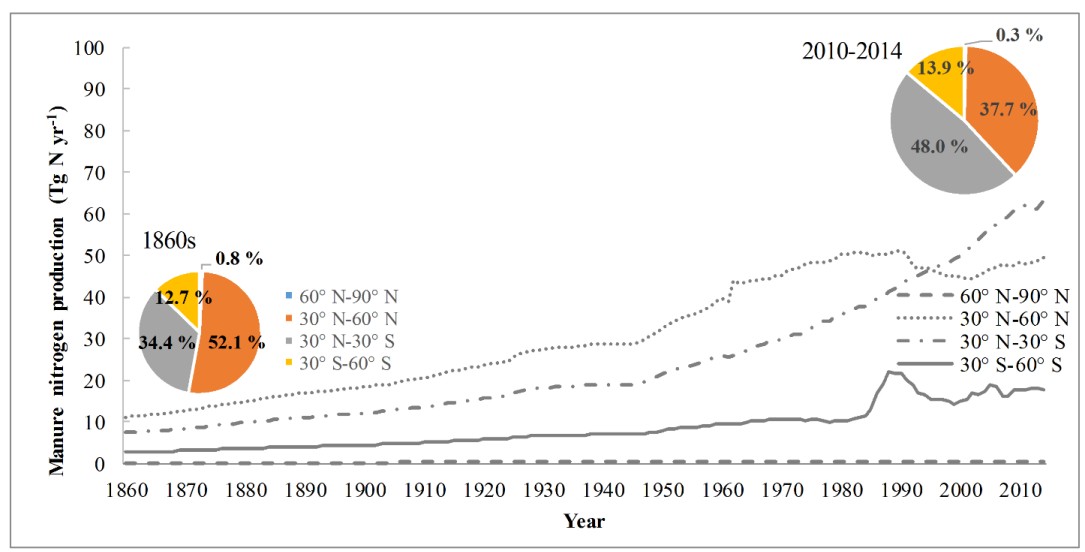


**Figure 4 Estimation of global manure nitrogen production at Northern High-latitude (60° N-90° N),**
**Northern Mid-latitude (30° N-60° N), tropical region (30° N-30° S) and Southern Mid-latitude (30°**
**S-60° S)**



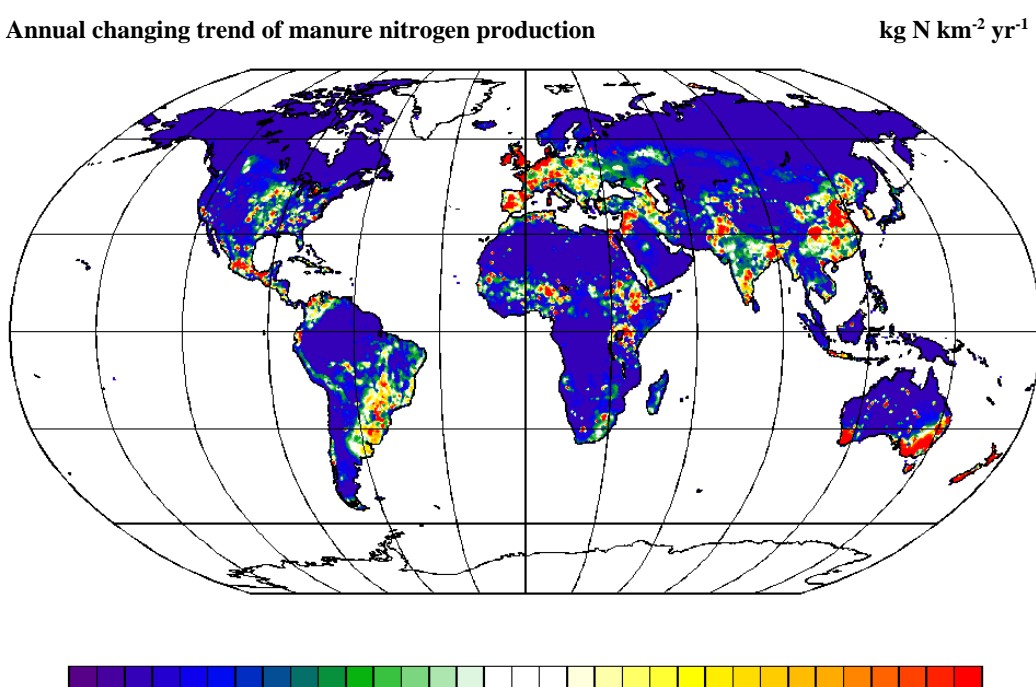

**Figure 5 Spatial variation in the annual changing trend of manure nitrogen production (kg N km$^{-2}$ yr$^{-1}$) during 1860-2014**





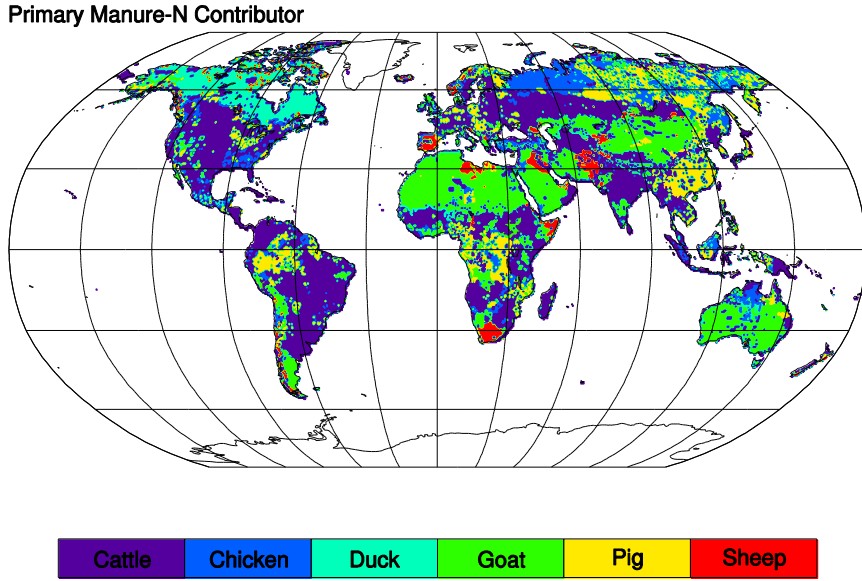


**Figure 6 Spatial distribution of the primary contributors to manure nitrogen in the year 2014**




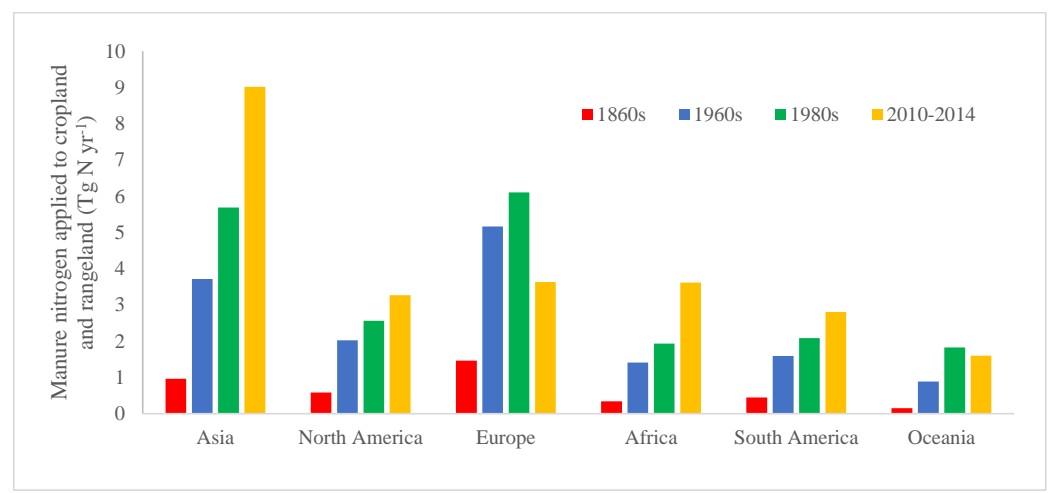


**Figure 7 Changes of manure nitrogen amount applied to cropland and rangeland at the continental level**





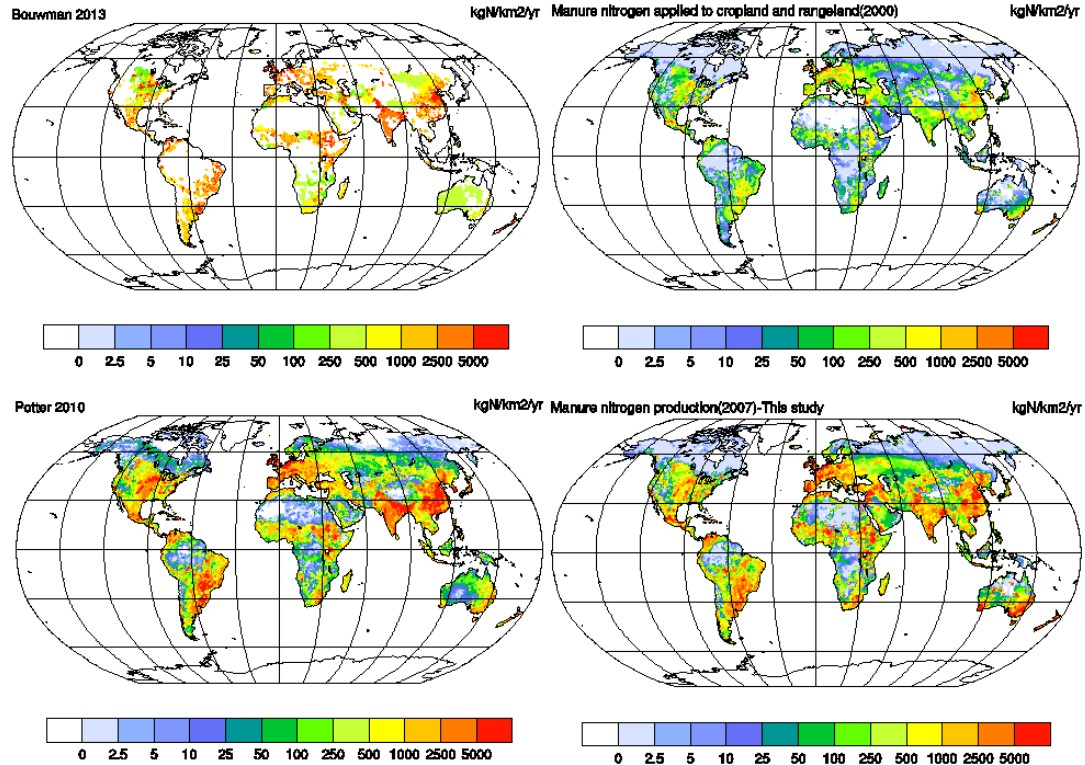


**Figure 8 Comparison of manure nitrogen production estimated by Bouwman et al., 2013, Potter et al., 2010, and this study**
