# Peer review of "Manure nitrogen production and application in cropland and rangeland during 1860 - 2014: A 5-minute gridded global data set for Earth system modeling"

_Earth System Science Data, 2017_

## Referee Comment (RC1) · Anonymous Referee #1 · 17 May 2017

Dear authors

The manuscript "Manure nitrogen production and application in cropland and rangeland during 1860–2014: A 5-minute gridded global data set for Earth system modeling" could be interesting for the broader communities including earth system modeling, climate scientist, and so on. Authors tried to make dataset for global manure inputs and their compositions i.e., cow, chicken, and the other animals manures. I agree this information is quite important information to assess historical global N cycling espe-

cially stimulated by anthropogenic activities. The quality of manures (i.e., CN ratio) are quite different among the animals and must affect global N cycling in environment. So, I think this paper and dataset is worth to publish in this journal. However, some important information is still lacking in this manuscript. So, I cannot recommend acceptance for the current manuscript.

**Major comments**

I don't have any strong objection for this data processing and the products. I think there are no perfect way to make such dataset. However, some important information are insufficient for methodology and results. For example, we can hardly follow the values used in this study. So, a substantial revision is needed to make this manuscript suitable for publication.

Another concerns are as follow;

**this is optional. not review comments** If you have already made "Carbon input" in manure application, could you share this dataset. This is also important information in GHG budget and nutrient cycling of cropland (and pasture).

**Individual comments**

**L97–99** Before this sentence, please add the detail explanation for GLIMS. For example; "what kind of dataset?", "how to make?", "What is the original dataset?".

**L97–99** Please add the citation for GLIMS.

**ESSDD**

**L108–111** The cited webpage said "Please mention the GLW version number with your citation.". Please clarify the version of dataset used in this study.

**L113** "global" → "national"?

**L117–118** For each livestock?

**L122–123** Are these values same among regions and periods?

**L125** Please clarify what kind of dataset are used in Holland et al. (2005) (N production? population?). Also, please add short description of this dataset.

**L130** How to deal missing data in FAOSTAT. Are there no missing data for total heads?

**L130–133** Before the this calculation, were GLIMS dataset (gridded data) aggregated to country based values?

**L130–133** What is the GLIMS's reference year (2005) in your study?

**L133** Is $k$ 1960–2014?

**L134** Please add the citation for IPCC 2006 guidelines.

**L136–142** A little bit tricky. Are there any information for country specific values of $Nrate_{i,j}$ (and $TAM_{i,j}$ in IPCC 2006 guidelines. So, please clarify "$i$" to be just China, US and the others.

**L136–142** Please clarify the values in $Nrate_{i,j}$ and $TAM_{i,j}$ used in this study, to show the differences among countries.

**L152–153** How to define the different agro-ecological areas.

**Eq.4, L154–168** Could you show all values used in this study? (if possible, as a table)

**L366–388** This sentence is not needed in the conclusion of this study.

**L175–179** Please show the values used in this study for each $F_M(j, ...)$.

**L175–179** Are there any reasons (or premises or citations) to use your assumed values? For example, you applied different values among different farming systems. Why? Also, I wonder why same values are applied throughout entire period (1860–2014?).

**L193–194** How about rangeland?

**L199–202** How to identify smallholder (or industry) in each grid? Please explain in detail.

**L237–244** I think the linear trend of manure production (i.e., kg N km$^{-2}$ yr$^{-1}$) is doubtful for this long time (1860–2014). This analysis should be separated to short periods (e.g., 1860–1910, 1910–1960, 1960–2010).

**L246–248** Please show the time-series data for fraction of livestock categories during 1960–2014.

**L3.4** Could you write the result of comparison with the other map (Fig./,8) in this section?

**L286** Why "$\pm$"?

**L301** Please specify → "during the past six decades"

**L304–312** These sentences are appropriate for Introduction. Please move to Intro or remove.

**L314–321** Please clarify this statistical analysis in the material & methods. Please keep in mind "correlation does not imply causation". Of course, I agree that manure
is important source of atmospheric $N_2O$. Inductive approach needs more careful logic and discussion.

**L323–326** Please remove this sentence.

**L332–334** Please clarify how large variance the excretion rate are exsting among regions (seasons)? Please cite previous studies for this value.

**4.3** Could you suggest the potential way how to reduce the uncertainties in this dataset?

**4.4** There seems to be no information and relevances to your dataset in this section. Please remove this or move to introduction.

**Fig 1** Honestly, from this figure, I can hardly follow data processing of this study (e.g., I cannot understand what the main dataset is and what is your data treatment). Perhaps, it is easy to understand using summary table for the dataset (i.e. data source, dataset, units, referreces).

**Fig 6** Please use "Swine" instead of "Pig".

---

## Editor Comment (EC1) · D. J. Carlson (Editor) · 17 May 2017

We have a careful review for this data set - many thanks to reviewer 1.

We have a second review ready, but that review references unpublished (e.g. confidential) material from another journal. We will close this discussion, add that second review, and ask authors to respond. Once the information contained in the second review becomes open (published), we will re-open this entire discussion for inclusion with the manuscript.

I appreciate cooperation of publisher, authors and reviewers as we manage these dif-

ferent policies among journals.

Dave Carlson

---

## Author Comment (AC1) · 15 Jun 2017

The manuscript "Manure nitrogen production and application in cropland and range-land during 1860–2014: A 5-minute gridded global data set for Earth system modeling" could be interesting for the broader communities including earth system modeling, climate scientist, and so on. Authors tried to make dataset for global manure inputs and their compositions i.e., cow, chicken, and the other animals manures. I agree this information is quite important information to assess historical global N cycling especially stimulated by anthropogenic activities. The quality of manures (i.e., CN ratio) are

quite different among the animals and must affect global N cycling in environment. So, I think this paper and dataset is worth to publish in this journal. However, some important information is still lacking in this manuscript. So, I cannot recommend acceptance for the current manuscript.

RESPONSE: Appreciate reviewer's positive comments on our study. We carefully addressed the questions raised by the reviewer and listed as below.

Major comments I don't have any strong objection for this data processing and the products. I think there are no perfect way to make such dataset. However, some important information are insufficient for methodology and results. For example, we can hardly follow the values used in this study. So, a substantial revision is needed to make this manuscript suitable for publication.

RESPONSE: Thanks for the precious comments. We have added the supplementary information (Table S1-3) to list most important parameters we used in this study in detailed.

Another concerns are as follow. this is optional. not review comments If you have already made "Carbon input" in manure application, could you share this dataset. This is also important information in GHG budget and nutrient cycling of cropland (and pasture).

RESPONSE: Thanks for the reviewer's interest in the related work. Unfortunately, we don't have carbon input from manure application. The current method (including equations and parameters) in this study is based on IPCC 2006 (Tie I), which was designed to calculate "the average nitrogen excretion rates" rather than carbon input. Therefore, we don't have carbon input from manure application. Individual comments

L97–99 Before this sentence, please add the detail explanation for GLIMS. For example; "what kind of dataset?", "how to make?", "What is the original dataset?".

RESPONSE: Thanks for reviewer's suggestions. We added the detailed explanation
for GLIMS. "The original Gridded Livestock of the World (GLW) database (Wint and Robinson, 2007), was further revised and improved in numerous ways, including collection of more up-to date livestock statistics, application of finer resolution predictor variables, and more reasonable analytical procedure, to develop the Global Livestock Impact Mapping System (GLIMS) (Robinson et al., 2014)."

L97–99 Please add the citation for GLIMS.

RESPONSE: Added.

L108–111 The cited webpage said "Please mention the GLW version number with your citation.". Please clarify the version of dataset used in this study

RESPONSE: Thanks for reviewer's suggestions. The version we used in this study is "GLW 2". We have added in the method part.

L113 "global" → "national"?

RESPONSE: We changed into "national".

L117–118 For each livestock?

RESPONSE: We changed livestock into "cattle".

L122–123 Are these values same among regions and periods?

RESPONSE: We revised the original sentences into "The annual trend was interpolated based on the five time periods (1960, 1970, 1980, 1990 and 1998) of livestock populations from the HYDE (History Database of the Global Environment, http://themasites.pbl.nl/tridion/en/themasites/hyde/landusedata/livestock/index-2.html) (Table S1). We applied those trends to fill the gaps for years without livestock populations from FAOSTAT" and added the table S1 to show the values from different regions and periods in more details.

L125 Please clarify what kind of dataset are used in Holland et al. (2005) (N production? population?). Also, please add short description of this dataset.

RESPONSE: By using the livestock population (including cattle, swine, sheep and poultry) from FAO Production Yearbooks, Holland et al. (2005) applied the default excretion rate suggested by Souchu and Etchanchu (1989) to generate the annual manure nitrogen production from 1860 to 1960. We obtained the trend of the inter-annual variation for manure nitrogen production before 1960 from Holland et al. (2005), and applied to each grid to estimate the amount of manure nitrogen production from 1860 to 1960.

L130 How to deal missing data in FAOSTAT. Are there no missing data for total heads?

RESPONSE: Yes, there are some missing data for livestock population from FAOSTAT. In that case, we applied the annual trend, which was interpolated based on the five time periods (1960, 1970, 1980, 1990 and 1998) of livestock populations from the HYDE (History Database of the Global Environment http://themasites.pbl.nl/tridion/en/themasites/hyde/landusedata/livestock/index-2.html) (Table S1), to fill the gaps for years without livestock populations from FAOSTAT.

L130–133 Before this calculation, were GLIMS dataset (gridded data) aggregated to country based values?

RESPONSE: GLIMS provided the country level statistics for livestock population in the supplementary information (Table S5) in Robinson et al. (2014).

L130–133 What is the GLIMS's reference year (2005) in your study?

RESPONSE: The GLIMS's reference year in our study was 2006.

L133 Is k 1960–2014?

RESPONSE: k is 1961-2014.

L134 Please add the citation for IPCC 2006 guidelines.

RESPONSE: Thanks, added.

L136–142 A little bit tricky. Are there any information for country specific values of Nratei,j (and T AMi,j in IPCC 2006 guidelines. So, please clarify "i" to be just China, US and the others.

RESPONSE: Thanks. IPCC 2006 guidelines only provide the regional nitrogen excretion rate, including North America, Western Europe, Eastern Europe, Oceania, Latin America, Africa, Middle East and Asia. Thus, we clarified "i" as China, US, and other regional specific values.

L136–142 Please clarify the values in Nratei,j and TAMi,j used in this study, to show the differences among countries

RESPONSE: We listed detail information in Table S2.

L152–153 How to define the different agro-ecological areas.

RESPONSE: Herrero et al. (2013) defined different agro-ecological zone based on temperature and length of growing period (LGP), the number of days per year during which crop growth is possible: - Arid and semi-arid, LGP ≤ 180 days. - Humid and sub-humid, LGP > 180 days. - Tropical highlands or temperate. Temperate regions are defined as those with one month or more with monthly mean temperature, corrected to sea level, below 5°C. Tropical highlands are defined as those areas with a daily mean temperature, during the growing period, of between 5 and 20 °C. In his study, he provided gridded global livestock production system (See Supplementary Information from Herrero et al. (2013)- Figure S1).

Eq.4, L154–168 Could you show all values used in this study? (if possible, as a table)

RESPONSE: We listed all the parameters from Eq. 4 in Table S3.

L366–388 This sentence is not needed in the conclusion of this study.

RESPONSE: I guessed the reviewer mean line 366-368. We removed that sentence.

L175–179 Please show the values used in this study for each FM(j, ...).

RESPONSE: We listed the values in Table S3.

L175–179 Are there any reasons (or premises or citations) to use your assumed values? For example, you applied different values among different farming systems. Why? Also, I wonder why same values are applied throughout entire period (1860–2014?).

RESPONSE: By using the livestock population data from FAO and the method from IPCC 2006 -Tier 1, we got the manure nitrogen production over 1860-2014. Beyond this, scientific communities are more interested in the manure application to the cropland and rangeland when thinking about food security, water security and sustainable development of entire livestock sector. Thus, we further processed the data to generate gridded manure application in cropland and rangeland. The parameters we used, particularly for different livestock production system has been broadly used in different studies to estimate the greenhouse gas emissions (Gerber et al., 2016; Herrero et al., 2016) and to assess the trade-off and synergies between the grazing intensity and ecosystem (Petz et al., 2014). The reason we used the same values throughout the entire period is due to the lack of available time-varying values.

L193–194 How about rangeland?

RESPONSE: The global livestock production system provided the rangeland distribution in 2006 (Herrero et al., 2013). The spatial distribution of livestock production systems in 2006 serves as a baseline map to characterize the change of livestock production system during 1860-2005. We assumed the spatial distribution of livestock production system remained the same during 2006-2014. We assumed if the grid cell was identified as rangeland-based systems, the livestock production system remained the same during the study period. If the grid cell was identified as mixed rainfed farming systems, the percent change of livestock production system would be proportional to the changes of the cropland area in that grid cell backward from 2006, and the mixed rainfed farming systems was converted from rangeland-based system

L237–244 I think the linear trend of manure production (i.e., kg N km$-2$ yr$-1$ ) is

doubtful for this long time (1860–2014). This analysis should be separated to short periods (e.g., 1860–1910, 1910–1960, 1960–2010).

RESPONSE: We separated the original figure into three short periods as reviewer suggested. We also added addition description as follow "Changes in manure nitrogen production showed highly spatial variability and revealed several hotspots over the globe due to the imbalances of global economic development and population growth (Fig. 4). Western Europe experience an increase in annual changing trend of manure nitrogen production from 1860 to late 1980s and decline thereafter. Southern Mexico, Central America, Columbia, Southern Brazil, Southeast Australia and India show a continuing increasing trend for manure nitrogen production during 1860-2014. Western and Eastern Africa and Northeast China experienced an increase in manure nitrogen production during the recent decades."

L246–248 Please show the time-series data for fraction of livestock categories during 1960–2014.

RESPONSE: Added as Figure 6

L3.4 Could you write the result of comparison with the other map (Fig./,8) in this section?

RESPONSE: Thanks for reviewer's suggestion. The current "comparison with other studies", particularly for Fig. 8 not only included the comparison of our estimation of manure application to cropland and rangeland with Bouwman et al., 2013 (as suggested by reviewer to add into Section 3.4), but also the comparison of our estimation of total manure production with Potter et al., 2012. And we also compare our estimate with other studies. Thus, we feel that it may be more appropriate and organized in the current format.

L286 Why "$\pm$"?

RESPONSE: We compared the continental estimate of manure production with Potter

et al., 2010's result. For Asia, North America, Europe and South America, our estimate is larger, but for Africa and Europe, our estimate is smaller. Thus, we add "±".

L301 Please specify → "during the past six decades"

RESPONSE: Changed into "during 1961-2013".

L304–312 These sentences are appropriate for Introduction. Please move to Intro or remove.

RESPONSE: Removed.

L314–321 Please clarify this statistical analysis in the material & methods. Please keep in mind "correlation does not imply causation". Of course, I agree that manure is important source of atmospheric N2O. Inductive approach needs more careful logic and discussion.

RESPONSE: We agreed with the reviewer, correlation does not imply causation. To avoid misleading the readers, we removed the correlation between atmospheric N2O and manure production part.

L323–326 Please remove this sentence.

RESPONSE: Removed.

L332–334 Please clarify how large variance the excretion rate is existing among regions (seasons)? Please cite previous studies for this value.

RESPONSE: Previous studies suggested that the excretion rates varied across regions. For example, the excretion rates in China for dairy and other cattle were $0.19 \pm 0.04$ and $0.07 \pm 0.02$ kg animal-1 day-1, respectively (Ouyang et al., 2013). The N excretion factors for EU countries using the gross N excretions in the Nitrates Directives reports was 75-184 and 20-90 kg N animal-1 yr-1 for dairy and other cattle, respectively (Velthof et al., 2015). Rufino et al. (2014) also suggested that the excretion rate could varied due to changes in feed quality and availability. Unfortunately, they didn't

give an estimate of how large the variances are among seasons.

4.3 Could you suggest the potential way how to reduce the uncertainties in this dataset?

RESPONSE: Reducing the associated uncertainty seems straightforward but hard to accomplish at the current stage, which needs to have more available data for excretion rate and livestock systems at finer scale with temporal variation. In addition, system thinking is another way to unravel the complexity and meanwhile explore the option for the sustainable development (Section 4.3).

4.4 There seems to be no information and relevances to your dataset in this section. Please remove this or move to introduction.

RESPONSE: Removed.

Fig 1 Honestly, from this figure, I can hardly follow data processing of this study (e.g., I cannot understand what the main dataset is and what is your data treatment). Perhaps, it is easy to understand using summary table for the dataset (i.e. data source, dataset, units, referreces).

RESPONSE: We removed the original Figure 1 and added Table 1 as a summary table.

Fig 6 Please use "Swine" instead of "Pig".

RESPONSE: Revised.

Reference:

Gerber, J. S., Carlson, K. M., Makowski, D., Mueller, N. D., Garcia de Cortazar-Atauri, I., Havlík, P., Herrero, M., Launay, M., O'Connell, C. S., and Smith, P.: Spatially explicit estimates of N2O emissions from croplands suggest climate mitigation opportunities from improved fertilizer management, Global Change Biol, 2016. 2016.

Herrero, M., Havlík, P., Valin, H., Notenbaert, A., Rufino, M. C., Thornton, P. K., Blüm-mel, M., Weiss, F., Grace, D., and Obersteiner, M.: Biomass use, production, feed efficiencies, and greenhouse gas emissions from global livestock systems, Proceedings of the National Academy of Sciences, 110, 20888-20893, 2013.

Herrero, M., Henderson, B., Havlík, P., Thornton, P. K., Conant, R. T., Smith, P., Wirsenius, S., Hristov, A. N., Gerber, P., and Gill, M.: Greenhouse gas mitigation potentials in the livestock sector, Nature Climate Change, 2016.

Holland, E. A., Lee-Taylor, J., Nevison, C., and Sulzman, J.: Global N Cycle: Fluxes and N2O Mixing Ratios Originating from Human Activity. Data set. Available online [http://www.daac.ornl.gov] from Oak Ridge National Laboratory Distributed Active Archive Center, Oak Ridge, Tennessee, U.S.A. doi:10.3334/ORNLDAAC/797., 2005.

Ouyang, W., Hao, F., Wei, X., and Huang, H.: Spatial and temporal trend of Chinese manure nutrient pollution and assimilation capacity of cropland and grassland, Environmental Science and Pollution Research, 20, 5036-5046, 2013.

Petz, K., Alkemade, R., Bakkenes, M., Schulp, C. J., van der Velde, M., and Leemans, R.: Mapping and modelling trade-offs and synergies between grazing intensity and ecosystem services in rangelands using global-scale datasets and models, Global Environmental Change, 29, 223-234, 2014.

Robinson, T. P., Wint, G. R. W., Conchedda, G., Van Boeckel, T. P., Ercoli, V., Palamara, E., Cinardi, G., D'Aietti, L., Hay, S. I., and Gilbert, M.: Mapping the Global Distribution of Livestock, Plos One, 9, 2014. Rufino, M., Brandt, P., Herrero, M., and Butterbach-Bahl, K.: Reducing uncertainty in nitrogen budgets for African livestock systems, Environ Res Lett, 9, 105008, 2014.

Souchu, P. and Etchanchu, D.: The Environmental Effects of the Intensive Application of Nitrogen Fertilizers in Western Europe: Past Problems and Future Prospects, 1989. 1989.

[Figure]

Velthof, G. L., Hou, Y., and Oenema, O.: Nitrogen excretion factors of livestock in the European Union: a review, Journal of the Science of Food and Agriculture, 95, 3004-3014, 2015.

Wint, W. and Robinson, T.: Gridded livestock of the world, Food and Agriculture Organization of the United Nations, Rome. 131p, 2007. 2007.

Please also note the supplement to this comment:
http://www.earth-syst-sci-data-discuss.net/essd-2017-11/essd-2017-11-AC1-supplement.zip